# Influences of Spatial Accessibility and Service Capacity on the Utilization of Elderly-Care Facilities: A Case Study of the Main Urban Area of Chongqing

**DOI:** 10.3390/ijerph20064730

**Published:** 2023-03-08

**Authors:** Jinhui Ma, Haijing Huang, Daibin Liu

**Affiliations:** 1School of Architecture and Urban Planning, Chongqing University, Chongqing 400044, China; 2Key Laboratory of New Technology for Construction of Cities in Mountain Area, Chongqing University, Chongqing 400044, China; 3School of Economics, Guangxi Minzu University, Nanning 530008, China

**Keywords:** utilization, spatial accessibility, elderly-care facilities (ECFs), two-step floating catchment area method, service capacity

## Abstract

With the unprecedented growth of the elderly population in China, elderly-care facilities (ECFs) are in a fast expansion process. However, limited attention has been paid to the imbalance at the actual utilization level of ECFs. This research aims to reveal the spatial inequity of ECFs and to quantitatively examine the effect of accessibility and institutional service capacity on utilization. Taking Chongqing, China, as the study area, we measured the spatial accessibility of different travel modes by the Gaussian Two-Step Floating Catchment Area (G2SFCA) method and investigated distribution differences in spatial accessibility, service capacity, and utilization of ECFs by the Dagum Gini Coefficient and its decomposition. Then, the impact of spatial accessibility and service capacity on the utilization of regional ECFs was quantified by multiscale geographically weighted regression (MGWR). The study findings can be summarized as follows. (1) Walking accessibility has the most significant impact on the utilization of ECFs and shows geographic heterogeneity. Developing a pedestrian-oriented network of pathways is essential to enhance the utilization of ECFs. (2) Accessibility by driving and bus-riding does not correlate with regional ECFs utilization, and relevant studies cannot rely on them alone for assessing the equity of ECFs. (3) In the utilization of ECFs, since the inter-regional difference is more significant than the intra-regional difference, efforts to reduce the overall imbalance should be oriented toward inter-regional variation. The study’s findings will assist national policymakers in developing EFCs to enhance health indicators and quality of life for older adults by prioritizing financing for shortage areas, coordinating ECFs services, and optimizing road systems.

## 1. Introduction

Population aging is growing increasingly significant and has become a major social issue worldwide [1]. According to the United Nations report, there were 703 million older persons (aged 65 and over) globally in 2019, with the number expected to double in the following three decades to reach 1.5 billion by 2050 [2]. The growing elderly population poses a huge challenge to social development [3], especially in developing countries [4]. China has the most older adults globally and is also one of the countries with the fastest ageing population [5]. According to the latest 7th Census of China in 2020, the old population (aged 60 and over) reached 264 million, accounting for 18.7% of the total population, a 5.44 percentage point rise from the sixth census [6]. This number is predicted to reach 30% by 2050 [7]. As the population ages, China will continue to face the pressure of long-term balanced population development [8].

The physical condition of the elderly deteriorates with age [9], which increases their demand for elderly-care facilities (ECFs) [10]. Since 2012, China has experienced a tremendous expansion in the number of aged-care facilities, particularly in megacities where ageing is a significant issue [11]. Local governments, such as in Beijing, Shanghai and Sichuan, place a high value on the ageing population [12]. Researchers discovered that, despite increasing service facilities, supply still falls short of meeting demand [13]. By 2020, China’s ECFs were only able to service 3.1% of the elderly population [14]. Due to limited urban space resources and rising demand for ECFs, the inequitable allocation of resources for urban services has increased. In China, the geographical scope of the studies has been uneven. Many studies examined the inequity of service of ECFs in plain cities such as Beijing, Tianjin and Shanghai. However, few examined Chongqing, a mountainous city with a unique road network. With the largest elderly population in China, Chongqing is experiencing severe ageing.

The inequality of social service resources for disadvantaged groups has been a primary concern in sustainable urban development [15,16]. Researchers have grown more interested in EFC disparity in recent years [17]. Various methods have been used to assess public service equality, including spatial accessibility, Gini coefficient analysis, and spatial autocorrelation analysis [3]. Spatial accessibility is commonly used to evaluate people’s ease of access to public services at a specific location [18]. The main methods for evaluating the spatial accessibility of public service facilities are the Huff model [19], kernel density method [20], gravity model method [21] and two-step moving search method [22]. Compared with other methods, the two-step moving search method is more accurate and comprehensive in measuring the spatial accessibility of facilities and has been more widely developed and applied [23]. To improve the study’s rationale and application, several academics have expanded on the 2SFCA model by adding distance decay functions [24], variable catchment area [25], traffic mode categorization [26], and inter-service facility competitive effects [27]. The Gini coefficient is a global statistic that measures the income disparity of a country’s or region’s population [28]. It is commonly used to analyze the balance of socioeconomic indicators among various population groups [29], and is applied in various fields, including biology, economics, and transportation [30]. Furthermore, spatial autocorrelation analysis is a valuable method for studying the degree of geographical correlation and dependency of specific variables in analyzing the equity of municipal facility services [31].

Although some studies on the application of the equity method enhancement and assessment have been done, the existing studies still have some limitations. First, there is not only a quantitative imbalance in the supply of, and demand for, ECFs. More seriously, there is also an imbalance in space utilization [10], which has been ignored in prior studies. This imbalance in utilization is mainly reflected through the actual access to ECFs services by regional older adults, i.e., it shows the degree of acceptance of ECFs services. In recent years, there has been an unusual phenomenon in China: older adults in city centres have been waiting in line for ECFs for months or more (named “a bed is hard to acquire” in China), yet in remote suburbs, the opposite is true. This issue suggests that accessibility indicators only may not accurately reflect the equality of the spatial distribution of ECFs. Second, it is generally accepted that studies of spatial accessibility based on quantitative relationships between supply and demand can represent the equality of facility services to some extent. However, few studies have investigated the influence and relationship between spatial accessibility (supply and demand) and utilization (natural choice) in ECFs [32]. Finally, institutional service capability is an essential attribute of ECFs [33] and significantly influences the willingness to choose. It is worth exploring to what extent it affects the utilization of ECFs compared to spatial accessibility. Therefore, the main inquiry questions for this study are:What are the characteristics of the spatial distribution of ECF utilization, accessibility, and service capacity? In what areas do they exhibit significant imbalances?What are the main sources of the overall imbalance in the utilization of ECFs?How do accessibility and service capacity impact ECF utilization? What is the extent of the impact?

The purpose of this research is to reveal the spatial inequity of ECFs in Chongqing and to quantitatively examine the effect of accessibility and institutional service capacity on utilization. Furthermore, it provides a solid foundation for national policymakers to rationalize resource allocation for ECF services and optimize the distribution of the ageing population. The research framework for this paper is shown in Figure 1.

## 2. Materials and Methodology

### 2.1. Study Area

Chongqing, located in the southwest of China’s heartland, is the economic centre of the upper Yangtze River area. With the largest elderly population in China, Chongqing has been entering a state of deep ageing society. According to the Chongqing Statistical Yearbook 2021 [34], Chongqing’s senior population has reached 5.7 million. Chongqing has a 17.8% ageing rate (people over the age of 65), with a 4.1% growth rate, which is 3.6% higher than the national average. Notably, the main urban area is 7% of the city’s area but hosts nearly one-third of the city’s ageing population. This makes the ageing problem in the main urban area of Chongqing even more prominent. Therefore, the scope of this paper focuses on the balanced distribution of elder care facilities resources in the main urban area of Chongqing. The study area includes Yuzhong District, Dadukou District, Jiangbei District, Shapingba District, Jiulongpo District, Nanan District, Beibei District, Yubei District and Banan District (Figure 2).

### 2.2. Data Sources and Preprocessing

#### 2.2.1. Elderly Population Data

To obtain a more accurate spatial analysis, we used the sub-district (jiedao), a form of township-level administrative division in China, as the basic spatial unit to measure the inequity of elderly care institutions in the main urban area of Chongqing. The seventh national census collected data on the senior population in Chongqing’s 161 sub-districts [6]. Some scholars calculate accessibility using the geometric centre of the sub-district or its administrative centre [35]. The method does not consider the heterogeneity of the population distribution within a sub-district and may cause some errors. Therefore, the weighted centre of the elderly population has been used as the population centre of the sub-district in this paper.

#### 2.2.2. Elderly-Care Facilities Data

To ensure consistency between geographical and demographic data, we extracted information from the Chongqing Civil Affairs Bureau about elderly institutions in 2021 [36]. A total of 356 ECFs were identified in the main urban area of Chongqing during the study period. In the database, each record contains the following information: the name of the institution, the address, the number of elderly beds, the number of senior residents, and the number of people it serves. Table 1 shows the results of the partition statistics. According to the Chinese National Standard “Classification and Evaluation of Aged Care Institutions” [37], ECFs are divided into three categories: large (≥ 300 beds), medium (100–300 beds) and small (<100 beds). First, we determined the spatial latitude and longitude using the Map Coordinate Picker tool based on the addresses of the ECFs. Second, the coordinates were converted into point data using ArcGIS and overlaid with the Chongqing sub-districts layer. Finally, the spatial corrections were carried out by identifying the sub-districts in which the institutions were located. We quantified the utilization of ECFs by the ratio of actual number of elderly people in ECFs to the number of local older adults at the sub-district level. This more accurately reflects the equity of services at the regional level. Furthermore, to a large extent, the quality of services is determined by the availability of skilled healthcare staff [38]. To find out how many care staff provided by ECFs can be assigned to beds in each ECFs, service capacity is represented by the ratio of the number of nursing staffs by the number of beds.

#### 2.2.3. Network Data

The spatial data of the road network is derived from the Gaode Map and the Open Street Map, which mainly consists of the rail transit network, bus network, and urban roads (Figure 3). According to Yang et al. [39], the type of provider and the difference in transport costs can influence the size of the catchment area. Accordingly, based on a sample survey of Chinese older adults [40], the maximum travel time was set at 30, 60, and 90 min for small, medium, and large ECFs, respectively. Following the Road Traffic Safety Law of the People’s Republic of China [41], the standard speed for highways, trunk roads, secondary roads, and other roads is 120, 70, 40, and 30 km/h, respectively. In previous medical studies [42], walking speed was found to be 0.90 to 1.30 m/s for older people, so 3.5 km/h was determined as the average walking speed. Finally, according to the Chongqing Road Traffic Safety Regulations and the design code of the rail transit in Chongqing [43,44], the rail transit and the bus speed were set at 100 km/h and 50 km/h, respectively.

### 2.3. Accessibility Calculation

The traditional two-step floating catchment area method (2SFCA) treats supply and demand point interactions by dichotomy [45]. However, this method has two problems: the unreasonable setting of the catchment area, and the homogeneity of weight in the search domain. Researchers have suggested enhancing the two-part moving search technique by using distance decay functions, including kernel density function [46], exponential function [47], power function [48] and Gaussian function [49]. The Gaussian two-step floating catchment area method (G2SFCA) replaces the 0 and 1 in the dichotomous technique with distance decay weights, making the computation more accurate. This study used G2SFCA to evaluate the accessibility of elderly care institutions in Chongqing based on the Gaussian decay function.

Step 1: Calculating of the ratio of beds to elderly population in the catchment area of the ECF *j*:(1)$$R_{j}^{m}=\frac{S_{j}^{m}}{\sum_{k\in \left(d_{kj}\leq d_{o}^{m}\right)}D_{k}G\left(d_{kj},d_{o}^{m}\right)}$$
where *m* is the class type of the ECFs, $R_{j}^{m}$ is the supply-to-demand ratio of the ECF *j* within the search threshold, $S_{j}^{m}$ represents the number of beds provided by the ECF *j*, $D_{k}$ represent the number of elderly people in the demand centre *k*, $d_{o}^{m}$ represents the the travel cost thresholds for m-type ECFs, $d_{kj}$ is the spatial distance between demand centre *k* and the ECF *j*, and $G\left(d_{kj},d_{o}^{m}\right)$ is Gaussian-weighted distance decay function, which can be expressed as Equation (2):(2)$$G\left(d_{kj},d_{o}^{m}\right)=\left\{\;\begin{array}{c} \frac{e^{-\frac{1}{2}\times {\left(\frac{d_{kj}}{d_{o}^{m}}\right)}^{2}}-e^{-\frac{1}{2}}}{1-e^{-\frac{1}{2}}}\;\;\;\;\;\;\;\;\;\;\;\;\;if\;d_{kj}\leq d_{o}^{m}\\ 0\;\;\;\;\;\;\;\;\;\;\;\;\;\;\;\;\;\;\;\;\;\;\;\;\;\;\;\;\;\;\;\;\;\;\;\;\;\;\;\;\;\;\;\;if\;d_{kj}\leq d_{o}^{m} \end{array}\right.\;$$

Step 2: calculating the spatial accessibility index $A_{i}$ for demand centre i. The formulation is denoted as Equations (3) and (4):(3)$$A_{i}^{m}=\underset{k\in \left(d_{kj}\leq d_{o}^{m}\right)}{\sum }R_{j}^{m}G\left(d_{kj},d_{o}^{m}\right)$$
(4)$$A_{i}=\underset{m=1}{\sum }A_{i}^{m}$$
where $A_{i}^{m}$ is the spatial accessibility in the demand centre i to the m-type ECF and $A_{i}$ is the spatial accessibility in the demand centre *i* to the ECFs, calculated by summing the accessibility of the three types of ECFs.

### 2.4. Inequity Calculation

We use the Dagum Gini decomposition method to measure regional differences. This method overcomes the limitations of the traditional Gini coefficient and the Thiel index, and effectively analyzes the causes of regional differences [50]. Additionally, the method accurately decomposes the net interregional difference contribution to the overall regional difference by resolving the cross-over problem between subgroups.

The overall Gini coefficient is calculated as follows:(5)$$G=\frac{\sum_{j=1}^{k}\sum_{h=1}^{k}\sum_{i=1}^{n_{j}}\sum_{r=1}^{n_{h}}\left|y_{ji}-y_{hr}\right|}{2\mu n^{2}}$$
where G is the overall Gini coefficient, $y_{ji}\left(y_{hr}\right)$ is the variable value of a sub-district in the district jh, μ is the mean variable value of all sub-districts in the mean urban area of Chongqing, n is the total number of sub-districts, k is the number of districts, and $n_{j}\left(n_{h}\right)$ is the number of sub-districts in the district $j\left(h\right)$.

The overall Gini coefficient G can be decomposed into three parts. (i) Intra-regional difference *G_w_*, i.e., the spatial differences between variable value within districts, which refers to such differences between sub-districts within nine districts in this study. (ii) Inter-regional difference *G_nb_*, i.e., the differences of variable value between districts, which refers to such differences between nine districts in this study. iii) Intensity of transvariation *G_t_*, i.e., the differences of variable value between districts, which refers to such differences between nine districts in this study, refers to the remainder of the overall Gini coefficient generated by the overlapping effects between regions. The relationship between them satisfies  G *= G_w_ + G_nb_ + G_t_*. The formula is as follows:(6)$$G_{w}=\overset{k}{\underset{j=1}{\sum }}G_{jj}P_{j}S_{j}$$
(7)$$G_{jj}=\frac{\sum_{i=1}^{n_{j}}\sum_{r=1}^{n_{h}}\left|y_{ji}-y_{jr}\right|}{2\bar{Y_{j}}n_{j}^{2}}$$
(8)$$G_{nb}=\overset{k}{\underset{j=2}{\sum }}\overset{j-1}{\underset{h=2}{\sum }}G_{jh}(p_{j}s_{h}+p_{h}s_{j})D_{jh}$$
(9)$$G_{jh}=\frac{\sum_{i=1}^{n_{j}}\sum_{r=1}^{n_{k}}\left|y_{ji}-y_{jr}\right|}{n_{j}n_{h}\left(\bar{Y_{j}}+\bar{Y_{h}}\right)}$$
(10)$$G_{t}=\overset{k}{\underset{j=2}{\sum }}\overset{j-1}{\underset{h=1}{\sum }}G_{jh}(p_{j}s_{h}+p_{h}s_{j})(1-D_{jh})$$
(11)$$D_{jh\;}=\frac{d_{jh}-p_{jh}}{d_{jh}+p_{jh}}$$
(12)$$d_{jh}=\int_{0}^{\infty }dF_{j}\left(y\right)\int_{0}^{y}\left(y-x\right)dF_{h}\left(x\right);$$
(13)$$p_{jh}=\int_{0}^{\infty }dF_{h}\left(y\right)\int_{0}^{y}\left(y-x\right)dF_{j}\left(x\right)$$
where $P_{j}=n_{j}/n$, $s_{j}=n_{j}\bar{Y_{j}}/n\bar{Y}$. $D_{jh}$ represents the interaction between the variables in regions *j* and *h*, $d_{jh}$ is the difference in the variables between regions *j* and *h*, $p_{jh}$ is the first moment of transvariation, and $F_{h}$ and $F_{j}$ are the cumulative density distribution functions of regions *j* and *h* respectively.

### 2.5. Regression Model

The traditional OLS is a clear and simple linear regression. However, it ignores changes in spatial variables caused by fluctuating spatial locations. Researcher have offered geographically weighted regression (GWR) as an excellent solution to the problem of geographical non-smoothness using regional regression analysis with variable parameters [51]. Unlike the traditional GWR model, the multiscale geographically weighted regression model (MGWR) calculates the bandwidth of the explanatory variables, which improves the accuracy of the regression results [52]. Hence, we use MGWR to investigate the influence of institutional service capability and spatial accessibility on EFC utilization at various subdistrict levels. Its equation is as follows:(14)$$y_{i}=\beta_{bw0}\left(u_{i},v_{i}\right)+\sum_{k=1}^{m}\beta_{bwk}\left(u_{i},v_{i}\right)x_{ik}+\varepsilon_{i}$$
where $y_{i}$ is the utilization of EFCs (response variable), bwk is the bandwidth, $\beta_{bw0}\left(u_{i},v_{i}\right)$ is the regression intercept of the bw0, $\beta_{bwk}\left(u_{i},v_{i}\right)$ is the regression coefficients for variable k at optimal bandwidth bwk, $x_{ik}$ is the value of the variable $x_{k}$ at observation point i, and $\varepsilon_{i}$ is the random error term.

## 3. Results

### 3.1. Spatial Accessibility of Elderly-Care Facilities by Travel Mode

The spatial accessibility of ECFs in different places shows whether older individuals have equal access to ECFs. To answer questions 1 and 2 mentioned in the introduction, we focused on two aspects: the spatial distribution characteristics of accessibility and the quantification of accessibility discrepancies.

#### 3.1.1. Spatial Accessibility Distribution Characteristics

The accessibility score is the number of ECF beds available to each older adult using a specific method of transportation, and is shown by its colour: redder indicates more accessibility, while yellower indicates lower accessibility. The accessibility distribution varies greatly between modes of transportation. Accessibility to ECFs by rail transit, car, and bus is presented in Figure 4a–c, respectively. High accessibility presents a spatial agglomeration centred on the Yuzhong District. The closer the sub-districts location to Yuzhong District, the higher the accessibility score. Yuzhong District is the centre of the main urban area of Chongqing, and its urban transportation system is significantly more developed than in other areas. Thus, these three modes of transportation to ECFs are poor for older adults in areas outside the city centre, such as Beibei, Banan, and Dadukou. Accessibility to ECFs by walking is similarly highly uneven, as shown in Figure 4d. However, unlike other modes of travel, sub-districts with relatively high accessibility by walking are mainly scattered and dotted. Meanwhile, these sub-districts are distributed far from the city centre.

#### 3.1.2. Decomposition of Regional Accessibility Differences

Table 2 illustrates the differences in accessibility to the main urban area of Chongqing at a global level based on the Dagum Gini coefficient. Among all modes of transportation, walking accessibility has the largest Gini coefficient, indicating that the spatial distribution of walking accessibility is the most unequal. In terms of intra-regional variation, by comparing *G_w_*, we identified the districts with the most uneven accessibility distributions at the sub-district level. Figure 5a–d shows the Gini coefficients for the main urban areas for rail transit, driving, bus-riding, and walking, respectively. Firstly, in terms of accessibility by rail transit (Figure 5a), Dadukou, Jiangbei, and Shapingba have relatively higher Gini coefficients, indicating that the spatial distribution of accessibility in these areas is relatively more inequitable. Secondly, according to Figure 5b, Ba’nan’s Gini coefficient for accessibility by driving is nearly twice that of the other regions. Then, among the main urban areas, Jiulongpo, Jiangbei, and Beibei have the most inequitable accessibility by bus (Figure 5c). Finally, in Figure 5d, we can see that Dadukou and Jiulongpo have the most uneven spatial distribution of walking accessibility.

From the perspective of inter-regional differences, we used *G_nb_* to identify districts with the greatest differences in accessibility between groups. As shown in Figure 6, inter-regional differences in accessibility by rail transit and walking are significantly larger, whereas inter-regional differences in accessibility by driving and bus riding are generally lower. Beibei and Ba’nan differ significantly from other regions in Figure 6a. The most significant inter-regional differences in walking accessibility were found in Yuzhong, Yubei, and Jiulongpo (Figure 6d). Compared to the other areas, these three have a greater density of older residents and are less accessible by walking.

### 3.2. Service Capacity and Utilization Rate of Elderly-Care Facilities

#### 3.2.1. Spatial Distribution Characteristics

Figure 7a shows the spatial distribution of the service capacity of ECFs in the main urban area of Chongqing. Some sub-districts do not have ECFs, so they are named after Nodata and expressed using slashes. Service capacity (the number of people served per bed) ranges from 0.03 to 0.73, with an unequal spatial distribution. The sub-districts with the highest service capacity resources have 20 times more resources than those with the lowest service capacity resources. Sub-districts closer to the city centre have a higher service capacity in Jiangbei, Jiulongpo and Yubei. However, there are still a few sub-districts remote from the city centre with high service capacity, such as those in northwestern Beibei, central Shapingba, and southern Dadukou. Thus, ECFs away from urban centres should continue to improve their service capacity.

Figure 7b shows the spatial distribution of ECFs utilization in the main urban area of Chongqing. Shapingba has the highest space utilization rate of 2.24% among the nine districts, while Jiangbei has the lowest rate of 0.52%. Furthermore, 36.7% of the sub-districts with a space utilization rate of less than 1% are distributed mainly around the city centre. Sub-districts with a space utilization rate of 1% to 2% account for 41.3% of the total and are primarily concentrated in Ba’nan. This disparity may be caused by older adults selecting ECFs using a cross-regional approach. Spatial distance is generally considered the most critical factor in selecting an ECF for older adults. However, studies have shown that service capacity is also an influential factor in the choice of ECFs for older people. Figure 7a,b shows that sub-districts with higher space utilization are relatively better served.

#### 3.2.2. Decomposition of Regional Differences

Table 3 illustrates that overall service capacity and spatial utilization are significantly imbalanced. From the comparison of contribution rates, this imbalance is primarily caused by inter-group differences *G_nb_* and intensity of transvariation *G_t_*. Therefore, identifying and reducing interregional differences is the first step toward ensuring equity in service capacity and spatial use in Chongqing. In addition, identifying and adjusting high imbalances within the region is also essential.

Based on the intra-regional differences *G_w_* shown in Figure 8, we can see that the region’s overall service capacity and spatial utilization of sub-districts show significant inequity. In Figure 8a, Ba’nan, Dadukou, and Jiulongpo all have *G_w_* values greater than 0.5. Regarding spatial utilization, the utilization rate of ECFs varies significantly between sub-districts in the Yuzhong, Nan’an and Dadukou regions. The *G_w_* value is exceptionally high in Yuzhong District, which reaches 0.8 (Figure 8b). As is shown in Figure 9, the overall inter-regional differences in service capacity and space utilization are significant, with *G_nb_* ranging from 0.37 to 0.66 and 0.33 to 0.76, respectively. In terms of service capacity, Banan, Yubei, and Jiulongpo have higher *G_nb_* values than the other districts (Figure 9a), so these three districts should be the main targets to decrease inter-regional variability. Space utilization in Yuzhong should receive attention (Figure 9b), as it reflects the most significant variability.

### 3.3. Impact of Accessibility and Service Capacity on the Utilization 

The MGWR model was used to analyze the overall and local effects of the factors on utilization. Table 4 provides descriptive statistics for the MGWR model regressions, including mean, standard deviation, median, maximum, bandwidth, and *p*-values. Comparing the t-values with the adjusted t-values (95%) indicates that accessibility by walking and rail transit, as well as service capacity, have a direct impact on the utilization of ECFs.

To explore the spatial heterogeneity of the impact of the three factors on utilization rates, we conducted a spatial mapping analysis of the correlation coefficients and *p*-values for each element in Figure 10. Figure 10a–c shows the correlation coefficients for rail transit, walking, and service capacity, respectively. Redder colors indicate a stronger correlation, while yellower colors indicate a weaker correlation. Figure 10d,e displays the *p*-values for rail transit, walking, and service capacity, with darker shades indicating statistically significant results. As shown in Figure 10a, accessibility by rail transit significantly affects the utilization of ECFs. Correlation coefficients ranged from 0.34 to 0.42. This suggests that rail transit development may provide older people access to ECFs. In the city centre and southwest area, accessibility by rail transit affects utilization most significantly, with *p*-values ranging from 0.40 to 0.42 (Figure 10d). Rail transit accessibility in the northeast has a less significant impact on utilization and is less significant. Because rail transit development has been concentrated in the centre and southwest, the northeast has limited access to this resource.

Compared with accessibility by rail transit, walking accessibility has a more significant spatial effect on the utilization of ECFs, with coefficients ranging from 0.22 to 1.13 (Figure 10b), decreasing stratified from west to east. In the western region (Yuzhong and Shapingba), walking accessibility positively affects utilization, with coefficients ranging from 0.74 to 1.13. In the central region (Dadukou, Jiangbei, Beibei and Yubei), walking accessibility has a relatively low impact on utilization but is still significant (Figure 10e). In the eastern region (Banan District), walking accessibility has the lowest impact on utilization and is insignificant, with coefficients ranging from 0.22 to 0.39. Additionally, there is geographic heterogeneity in the impact of service capacity on spatial utilization, but the degree of impact is relatively low overall, with coefficients ranging from 0.24 to 0.36 (Figure 10c). In the southwest region (Dadukou, Shapingba, and Jiulongpo districts), service capacity greatly influences spatial utilization. The correlation coefficient and significance level decrease from the southwest to the northeast (Figure 10f).

## 4. Discussion

### 4.1. Principal Findings and Comparison with Other Studies

Given the increasing inequality of ECFs, existing studies fail to provide evidence regarding the impact of accessibility and service capacity on the utilization of ECFs. Contrary to previous studies based on POI [45], the data in this study were sourced from the Chongqing government’s latest statistical data on ECFs in 2020, ensuring complete and scientific findings. The findings from this study suggest that accessibility by rail transit and walking, and service capacity, can affect the utilization of ECFs. Globally, walking accessibility has the most significant impact on space utilization, while at the local scale, these impacts are variable. The contribution of inter-regional difference to the utilization of ECFs was more significant than intra-regional difference and was the main source of overall difference.

First, walking accessibility has the most significant impact on space utilization. Our findings accord with recent studies that travel by walking and bicycling was associated with more frequent primary care visits than car travel [53]. Since Chongqing is a mountainous city resulting in inconvenient cycling for the elderly, bicycling was not considered in this study. In contrast to primary care, our research focuses on ECFs, which provide more complete services for the elderly. In addition, we used the MGWR method to generate correlation results not only at the global level but also at the local level through spatial mapping. At the local level, we found that the higher the population density of the region, the lower the walking accessibility. This is consistent with the results of Zhang et al. [22]. Although the road system in the city centre is more developed than in other areas, accessibility by walking is relatively low. The reason for this is the extremely high density of elderly people in the city center (Figure 3). It is estimated that the population density in Yuzhong District is 30 times higher than that in the other regions (southeast of Banan District). Despite the huge number of ECFs clustered in city centers, per-person service resources remain limited. These areas have better public transport networks, leading to the possibility that other modes of travel (such as bus-riding and rail transit) may be more popular than walking. In these areas, walkability significantly impacts space utilization, with *p*-values less than 0.01. This indicates that walkability has a more significant impact on space utilization in densely populated areas.

Previous studies have suggested that walking is the most popular mode of transportation for older adults, and benefits their overall health and well-being [17,54,55]. Furthermore, a survey of older adults in Chongqing showed that 67.3% walked to travel, and 78.9% travelled within a two-kilometre radius [56]. Walking is the primary mode of transportation for older people in other regions as well [22]. Because of their physiological condition, older people have a low level of mobility and a limited range of trips. According to research, older adults’ satisfaction with care services decreases as walking distance increases [57]. Thus, as a safe and affordable method of transportation, walking is especially suitable for older adults [58]. In summary, choosing an ECF closer to home can enhance older adults’ sense of belonging and well-being. Older adults prefer ECFs that are located within walking distance.

Second, accessibility by rail transit and service capacity can also affect the utilization of ECFs, although less significantly than walking accessibility. Spatial distance is generally considered the most critical factor in selecting an ECF for older adults. However, studies have shown that service capacity is also an influential factor in the choice of ECFs for older people [59]. Through Figure 7a,b, it is apparent that sub-districts with higher space utilization are relatively better served. Our findings accord with recent studies indicating that distance to home, care services, and rail accessibility are all significant factors influencing older adults’ choices of ECFs [10]. According to Habib et al. [60], older adults are more likely to travel by public transportation if the system is reliable, convenient, and comfortable. In Chongqing, a mountain city with high heat and humidity, the relatively poor bus infrastructure (congested roads, slow speeds, and lack of seats and covers at bus stops) could discourage older adults from travelling to ECFs [61]. In recent years, the rail transit system has developed rapidly, offering increased safety, convenience and comfort over bus-riding. This gives older adults a new travel option when considering ECFs. Since service capacity is an effective guarantee of the quality of life for older adults in ECFs [62], improved service facilities can significantly enhance the use of these facilities.

Third, the inter-regional differences between regions of the explanatory and explained variables are significantly more significant than the intra-regional differences. In previous studies, the Gini coefficient has been used to measure the equity of public transportation [30] and various types of nursing homes [3]. However, this only explains the overall level of inequity. Using the Gagum Gini coefficient, we identify areas of imbalance and the main sources of overall differences. Across the nine districts, we discovered various degrees of intra-regional and inter-regional differences, and this variation was intuitively reflected in the coefficient values. For districts with more pronounced intra-regional differences, we can reduce the imbalance directly at the sub-district level, such as the walking accessibility of Jiulongpo District (Figure 6d). For districts with more apparent inter-regional differences, resources can be rationally allocated across districts, such as the service capacity between Yubei-Banan districts (Figure 9a). This method provides a measure of equity identification at different levels.

Somewhat surprisingly, accessibility by driving and bus-riding was not significantly associated with the utilization of ECFs at the global level. While previous research has focused on improving accessibility methods [63] and fairness assessments [45] for ECFs in driving mode, our findings demonstrate that these conclusions may not be fully applicable in practice since this mode does not contribute to utilization.

As we know, only older adults with cars can use this mode of transportation [64]. Unlike some developed countries, older adults rely on cars for transportation [65]. In developing countries, older adults have fewer transportation options, especially with low car ownership, which increases their transportation barriers [66]. In these countries, public transport networks (such as bus-riding and rail transit) are at the core of urban public facilities. This may overlook the fact that older people travel predominantly by walking, thus undermining their access to public health services. As a recent study from a medium to low-income country has shown, the accessibility to non-communicable disease services decreased as age increased [67]. Furthermore, transportation academics regard older adults as a “transportation disadvantaged group” [68], with cognitive and functional decline eliminating driving alternatives for most elderly adults [69]. Accordingly, older adults in middle-income countries do not travel by car as their primary mode of transportation [53]. Travel by bus is relatively crowded and slow, making it difficult for elders to travel safely, particularly to distant ECFs. Thus, most older adults may not choose these two modes of transportation to ECFs. These theories are logically consistent with our findings that accessibility by driving and bus-riding had no significant impact on the utilization of ECFs.

### 4.2. Limitations

We investigated the spatial differentiation characteristics of the equity of ECFs in the main urban area of Chongqing, identified the main sources of inequity, and explored the effects of transportation accessibility and service capacity on utilization. However, there are limitations to our study. (1) Our study’s explanatory variables included many factors (transportation network, type of institution, spatial location, travel time, population, beds, and attendants). Nevertheless, income, air quality and other uncertain information, such as subjective preferences of older adults in different age groups, were not considered. In the future, these data can be supplemented and improved with government guidance, and their impact on space utilization can be further investigated by overlaying income data and air quality. (2) In addition to the number of people served to the number of beds ratio, service capacity also includes the quality of meals and the quantity of other equipment, such as televisions and air conditioners. Due to the large sample size, these data are not currently available. Future research could develop more quantitative statistics on these elements and use more comprehensive service metrics to evaluate the quality of ECF services. This will facilitate a more accurate analysis of its contribution to utilization.

### 4.3. Implications

At the local level, sub-districts that need to be improved can be divided into two categories: without ECFs and with low ECFs utilization (Figure 7 shows that 25% of the sub-districts in the main urban area of Chongqing do not have ECFs). For sub-districts with low utilization of ECFs, the cost of enhancing rail transit construction is high. Therefore, for these older ECFs, improving the pedestrian system should be the first priority, followed by increasing service capacity. For sub-districts without ECFs, new projects should be located near sites with convenient accessibility by walking and rail transit. For example, in Chongqing, ECFs should be located within 2000 m of residential communities to ensure efficient operation.

At the global level, the government should focus on reducing regional differences and provide policy and economic support to areas with low spatial utilization of ECFs. This will ensure overall equity. In addition, separate strategies should be developed for different districts, with guided improvement measures to reduce the waste of urban public resources. For example, in areas with limited transportation options, the government could plan senior-specific bus routes and urban bus rapid transit for ECFs.

Future studies on the utilization of ECFs should integrate multiple transportation modes while giving sufficient weight to walking and rail travel. Furthermore, in this study, accessibility by bus and driving had no effect on the utilization of ECFs. Therefore, a single mode of travel by bus or car should be carefully considered in studies assessing the equity of ECFs in similar cities, especially methodological improvement studies. Finally, due to geographical variances, the effects of the same factor on fairness may differ between locations. Accordingly, future research on the utilization of ECFs for more geographical regions should be done.

## 5. Conclusions

Our findings are as follows. (1) Overall, walking accessibility has the most significant impact on the utilization of ECFs, and there is spatial heterogeneity in this significance. Building a pedestrian-friendly pathway system is critical for improving ECF utilization. (2) Regarding the utilization of ECFs, the inter-regional difference is significantly higher than the intra-regional difference and is the primary source of overall difference. Eliminating overall differences must consider the sensible deployment of resources across districts. (3) Accessibility by driving and bus-riding does not correlate with the utilization of ECFs. Therefore, relying on them alone to evaluate the service equity of ECFs is inadequate, and this should be avoided in future studies. We emphasize the actual utilization of ECFs as a more equitable criterion than single accessibility.

As the elderly population continues to grow, it is increasingly urgent to eliminate inequalities in services for ECFs. Therefore, it is imperative to develop separate strategies for unbalanced areas, with policy and economic support. Furthermore, appropriate policies should be oriented toward inter-regional differences in order to maintain ECF sustainability through coordinated resource allocation. The findings contribute to identifying service inequities in ECFs on an overall and local level, and provide a valuable reference for planners to develop coping strategies in the urban regeneration process.

## Figures and Tables

**Figure 1 ijerph-20-04730-f001:**
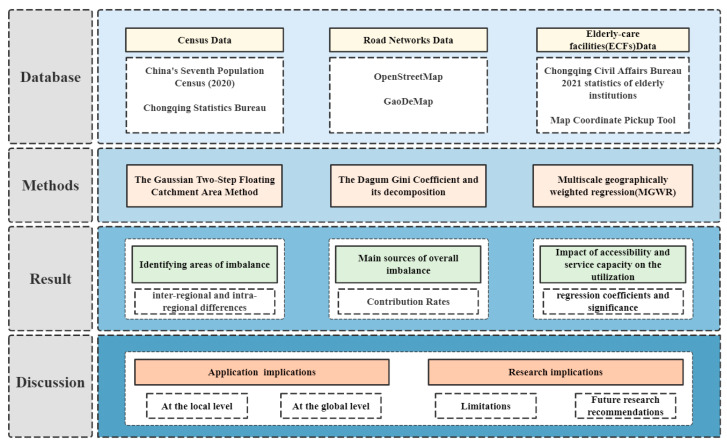
Research framework.

**Figure 2 ijerph-20-04730-f002:**
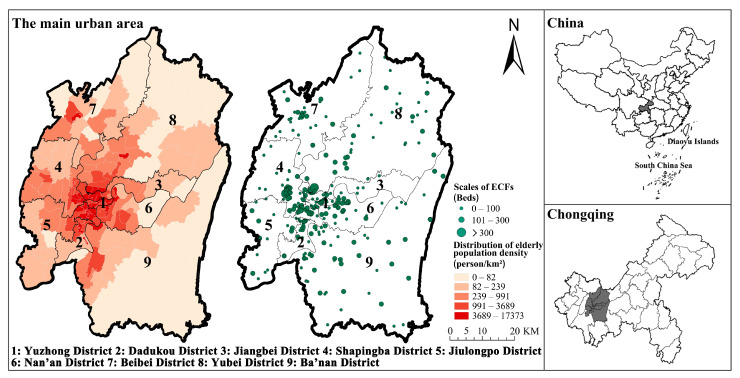
The geographical distribution of elder population density and ECFs in the main urban area of Chongqing.

**Figure 3 ijerph-20-04730-f003:**
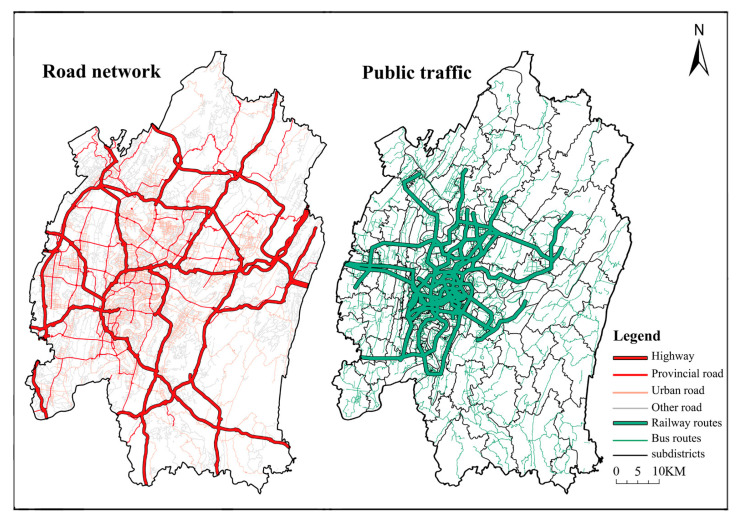
The location and Transport network of the main urban area of Chongqing.

**Figure 4 ijerph-20-04730-f004:**
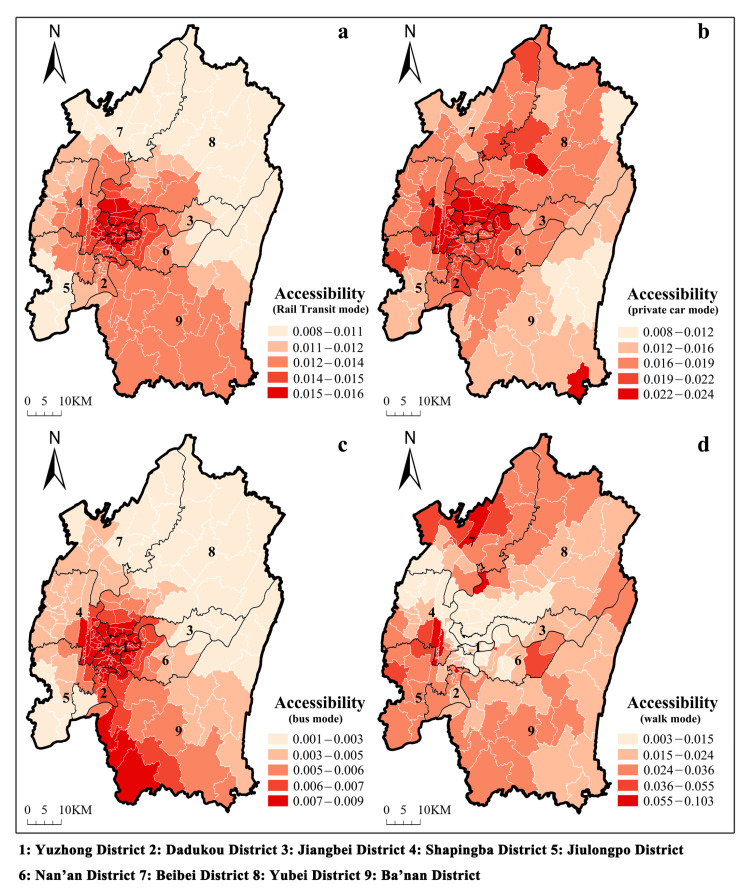
Spatial accessibility to ECFs under rail transit mode (**a**), private car mode (**b**), bus mode (**c**) and walk mode (**d**).

**Figure 5 ijerph-20-04730-f005:**
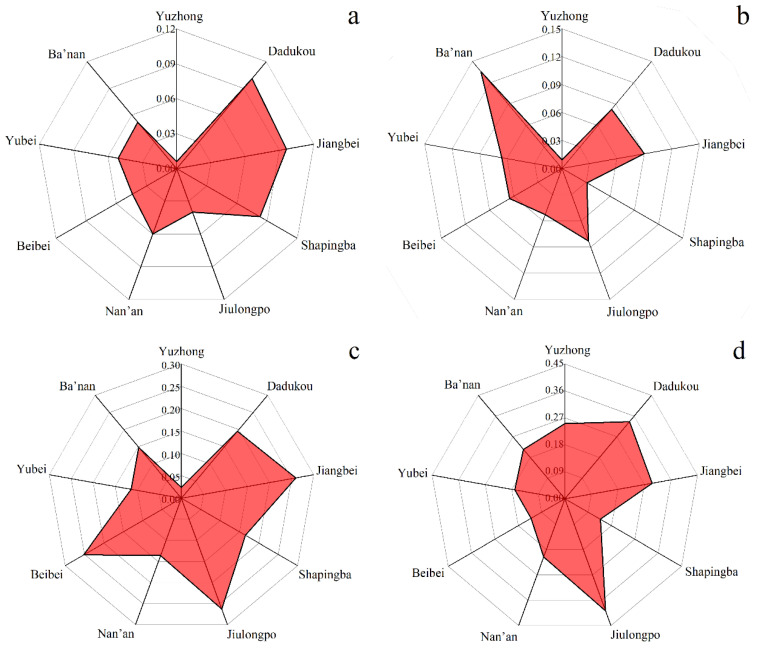
Intra-regional differences of The Dagum Gini coefficient. (**a**) Rail transit mode; (**b**) private car mode; (**c**) bus mode and (**d**) walk mode.

**Figure 6 ijerph-20-04730-f006:**
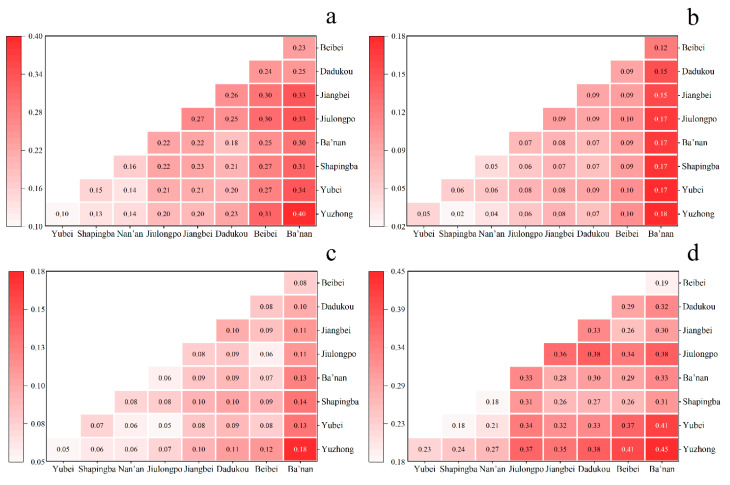
Inter-regional differences of The Dagum Gini coefficient. (**a**) Rail Transit mode; (**b**) private car mode; (**c**) bus mode and (**d**) walk mode.

**Figure 7 ijerph-20-04730-f007:**
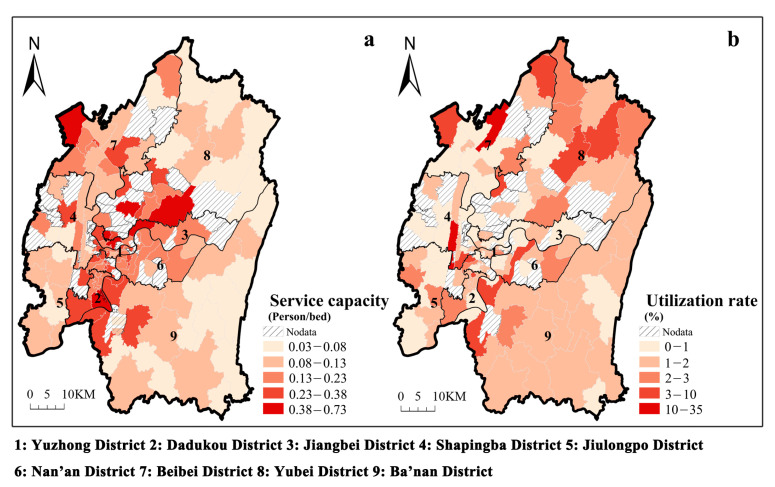
Spatial distribution of service capacity (**a**) and space utilization (**b**).

**Figure 8 ijerph-20-04730-f008:**
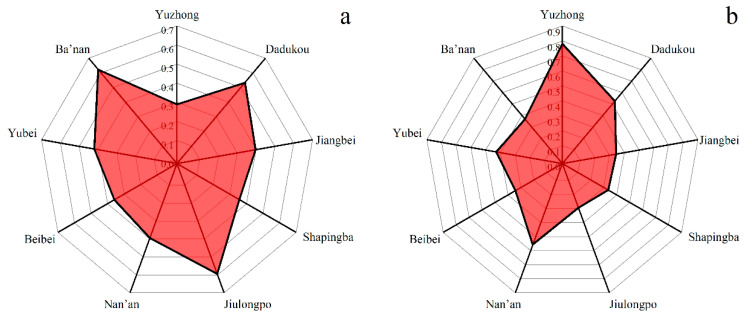
Intra-regional differences of The Dagum Gini coefficient. (**a**) service capacity; (**b**) space utilization.

**Figure 9 ijerph-20-04730-f009:**
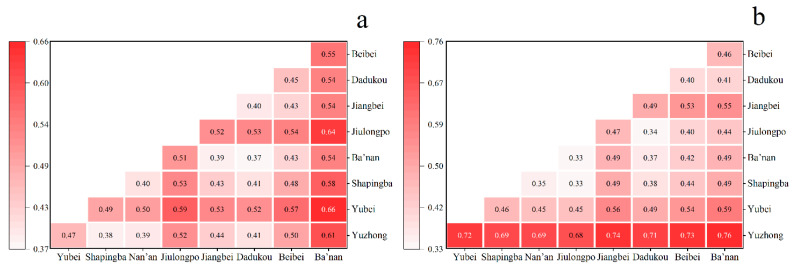
Inter-regional differences of The Dagum Gini coefficient. (**a**) service capacity; (**b**) space utilization.

**Figure 10 ijerph-20-04730-f010:**
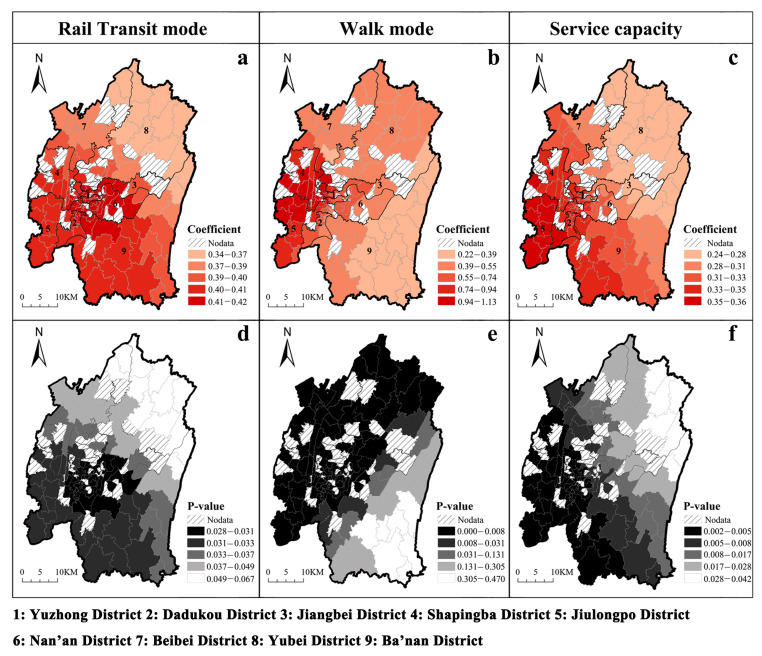
Spatial distribution of regression coefficients and significance.

**Table 1 ijerph-20-04730-t001:** EFCs of Different Districts.

District.	Area (km^2^)	Elderly Population (10,000 People)	Elderly-Care Facilities			
			No. of ECFs	No. of Occupants	No. of Care Staff	No. of Beds
Yuzhong	22.89	11.63	25	983	312	1586
Dadukou	103.81	8.26	31	1325	475	2280
Jiangbei	222.22	17.64	15	911	353	1304
Shapingba	396.72	24.79	74	5552	1193	8546
Jiulongpo	431.31	27.60	37	2746	748	3982
Nan’an	262.44	21.64	37	2448	693	4298
Beibei	753.56	17.72	37	2031	613	3413
Yubei	1454.62	35.94	42	3479	1027	4951
Ba’nan	1820.02	25.73	58	4874	1715	7851
Total	5467.59	190.97	356	24,349	7129	38,211

**Table 2 ijerph-20-04730-t002:** The Dagum Gini coefficient and its decomposition results.

Travel Mode	Overall	Contribution Rates (%)		
		G_w_	G_nb_	G_t_
Car	0.10	10.54	57.18	32.28
Subway	0.09	10.04	54.25	35.71
Bus	0.23	10.23	52.30	37.47
Walk	0.31	11.42	44.80	43.78

Notes: G_w_ refers to intra-regional differences; G_nb_ refers to inter-regional differences; G_t_ refers to intensity of transvariation.

**Table 3 ijerph-20-04730-t003:** The Dagum Gini coefficient and its decomposition results.

Type	Overall	Contribution Rates (%)		
		G_w_	G_nb_	G_t_
Service capacity	0.505	11.5665	29.2922	59.1414
Utilization rate	0.51092	11.5093	34.8646	53.6261

Notes: G_w_ refers to intra-regional differences; G_nb_ refers to inter-regional differences; G_t_ refers to intensity of transvariation.

**Table 4 ijerph-20-04730-t004:** Summary statistics for MGWR parameter estimates.

Variable	Mean	STD	Min	Median	Max	*p*-Value	Adjust t-Value (95%)	t
Intercept	−0.04	0.02	−0.07	−0.05	0.02	1.00	2.11	0.00
Rail transit accessibility	0.40	0.02	0.34	0.41	0.42	0.02	2.13	2.29
Car accessibility	−0.06	0.01	−0.08	−0.06	−0.04	0.75	2.18	0.32
Bus accessibility	−0.25	0.02	−0.31	−0.24	−0.23	0.07	2.14	−1.83
Walking accessibility	0.63	0.24	0.23	0.55	1.13	0.00	2.66	9.74
Service capacity	0.32	0.03	0.24	0.33	0.36	0.04	2.12	2.11

## Data Availability

The datasets used and/or analyzed during the current study are available from the corresponding author on reasonable request.

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
