# Peer review of "Influences of Spatial Accessibility and Service Capacity on the Utilization of Elderly-Care Facilities: A Case Study of the Main Urban Area of Chongqing"

_ijerph, 2023, doi:10.3390/ijerph20064730_

Round 1

Reviewer 1 Report

The article is a fascinating study examining inequality in access to health services among older people. Here are some observations that help to improve the article.

1. The text is very long; I suggest prioritising the most significant results. The authors could also remove tables and figures whose data already appear in the text. If they prefer to leave them, reduce the number of paragraphs where the same information appears.

2. In the Abstract, the authors say what they measure but do not say the article's objective. They also clearly state the results but not the conclusions. On the other hand, they also say what the conclusions are for but do not say what they are. I suggest adding these two elements to the Abstract (objective and conclusions).

3. In the Introduction, in line 37, are they talking about three decades or 30 years?

4. Clarify in line 42 whether the sixth census is the last to be carried out.

5. In line 91, they talk about "few studies". These studies should be cited.

6. Is the study based on hypotheses?

7. In the Method, in line 158, adjust the (300 > Beds ≥ 100).

8. In Figure 3, I suggest changing the Scales of ECFs (beds) colours. It is not easy to differentiate them if they are all the same colour. Putting circles with coloured borders and a transparent centre helps visually. You could also put two maps in the figure, one next to each other (as in Figure 4), one for the beds and one for the population distribution.

9. Putting the formula in the Method may be unnecessary because they mention the calculations they make in the text. However, they could move them to Supplementary if they prefer to leave the formulas.

10. In the Results section, only the findings should be included. The interpretation of the results should go into the Discussion. Adjust this in the paper. For example, this happens in lines 315-316: "The road network in these areas is relatively poorly developed" and 319-320: "Overall, reducing regional differences in accessibility should focus on improving accessibility by rail transit and walking in the corresponding areas." These explanations of possible causes and recommendations are given in the Discussion.

11. The footnotes to some figures, with the names of the districts, can go as columns 1 to 9 within the image.

12. Revise the beginning and end of lines 283 and 285 in the Results.

13. The number of figures should be reduced, leaving only the most relevant ones.

14. In Figure 11, the three greys-to-black scale maps do not allow for differentiating the darker shades.

15. In the Discussion section, ideas are repeated in other text parts. For example, in lines 416-418: "In this study, the G2SFCA method was used to calculate the accessibility of ECFs by different traffic modes, and the MGWR method was used to determine indicators related to utilisation". This has already been said.

16. The same is true in the Conclusions, in lines 535-539: "In this study, based on the database of ECFs provided by the Chongqing government, we applied Dagum Gini coefficient and its decomposition to identify areas of imbalance in supply-demand indicators and utilisation and to explore the main sources of overall imbalance. Then, we employed MGWR to quantitatively examine the effect of accessibility and institutional service capacity on utilisation and revealed the mechanisms of influence". This has already been said.

17. The purpose of the study should be at the beginning of the text, not in the Conclusions. Lines 549-551: "The purpose of this study is to identify regions of inequitable utilisation and investigate the major driving factors of it in order to develop more effective positive coping strategies".

18. I recommend reducing the number of bibliographies and citing only those texts that are directly related to the object of the study and that help to understand it better.

Author Response

Above all, thanks very much for the comments, which are very helpful to improve the quality of this article. We have revised the manuscript and especially paid much attention to your comments and suggestions.

Point 1: The text is very long; I suggest prioritising the most significant results. The authors could also remove tables and figures whose data already appear in the text. If they prefer to leave them, reduce the number of paragraphs where the same information appears.

Response 1: Thanks very much for your comments, which are very helpful to improve the quality of this article. We have removed the tables and figures whose data already appear in the text.

Point 2: In the Abstract, the authors say what they measure but do not say the article's objective. They also clearly state the results but not the conclusions. On the other hand, they also say what the conclusions are for but do not say what they are. I suggest adding these two elements to the Abstract (objective and conclusions).

Response 2: This comment is very valuable. The same suggestion has also been raised by other reviewers. We have revised the abstract to make the objective and conclusions of the study clearer(page 1 line 14-16ï¼›page 1 line 23-24).

Point 3: In the Introduction, in line 37, are they talking about three decades or 30 years?

Response 3: We are very sorry about our typos. This phrase has been modified to "three decades"(page 1 line 38).

Point 4: Clarify in line 42 whether the sixth census is the last to be carried out.

Response 4: Thank you for the suggestion. We have added the information" the latest 7th Census of China in 2020" required as explained above(page 1 line 41).

Point 5: In line 91, they talk about "few studies". These studies should be cited.

Response 5: We have cited these studies according to the reviewers' comments(page 2 line 94).

Point 6: Is the study based on hypotheses?

Response 6: Q1 and Q2 in the introduction are not based on assumptions. Q3 in our study is based on hypotheses. We formulated specific research questions and developed hypotheses to test these questions. These hypotheses guided our data collection, analysis, and interpretation of results.

Point 7: In the Method, in line 158, adjust the (300 > Beds ≥ 100).

Response 7: As suggested by the reviewer, we have adjust the “(300 > Beds ≥ 100)” into “(100-300 Beds)” (page 4 line 158).

Point 8: In Figure 3, I suggest changing the Scales of ECFs (beds) colours. It is not easy to differentiate them if they are all the same colour. Putting circles with coloured borders and a transparent centre helps visually. You could also put two maps in the figure, one next to each other (as in Figure 4), one for the beds and one for the population distribution.

Response 8: Thank you for your valuable suggestions, we have made changes to this figure. We put two maps in the figure, one next to each other, one for the beds and one for the population distribution (page 5 line 170).

Point 9: Putting the formula in the Method may be unnecessary because they mention the calculations they make in the text. However, they could move them to Supplementary if they prefer to leave the formulas.

Response 9: We completely understand the feedback from the reviewer and appreciate this comment. However, we believe that placing them in the methods section will not only enhance the reader's understanding but also ensure proper citation of the methods. In fact, this approach is commonly used in many scientific articles(DOI:10.3389/fpubh.2022.1003791, DOI:10.1016/j.puhe.2019.01.005).

Point 10: In the Results section, only the findings should be included. The interpretation of the results should go into the Discussion. Adjust this in the paper. For example, this happens in lines 315-316: "The road network in these areas is relatively poorly developed" and 319-320: "Overall, reducing regional differences in accessibility should focus on improving accessibility by rail transit and walking in the corresponding areas." These explanations of possible causes and recommendations are given in the Discussion.

Response 10: Thanks for your suggestion. We have moved these sentences to the DISCUSSION and made cuts to the repetitive points. Your suggestions have optimised the framework and logic of the thesis. These changes are marked in red in the revised paper.

Point 11: The footnotes to some figures, with the names of the districts, can go as columns 1 to 9 within the image.

Response 11: We have revised these figures(figure 2\4\7\10) according to the reviewers’ suggestion. (page 5 line 170, page 9 line 268, page 12 line 335, page 14 line 381).

Point 12: Revise the beginning and end of lines 283 and 285 in the Results.

Response 12: Thanks for your careful checks. We are sorry for our carelessness. Based on your comments, we have made the correction to make the sentence complete(page8 line 169).

Point 13: The number of figures should be reduced, leaving only the most relevant ones.

Response 13: As suggested by the reviewer, we have removed some figures and left the most important ones.

Point 14: In Figure 11, the three greys-to-black scale maps do not allow for differentiating the darker shades.

Response 14: We have revised this figure in more distinguishable colours according to the reviewers’ suggestion.

Point 15: In the Discussion section, ideas are repeated in other text parts. For example, in lines 416-418: "In this study, the G2SFCA method was used to calculate the accessibility of ECFs by different traffic modes, and the MGWR method was used to determine indicators related to utilisation". This has already been said.

Response 15: We have removed these duplicate texts according to the reviewers’ suggestion, in the DISCUSSION section.

Point 16: The same is true in the Conclusions, in lines 535-539: "In this study, based on the database of ECFs provided by the Chongqing government, we applied Dagum Gini coefficient and its decomposition to identify areas of imbalance in supply-demand indicators and utilisation and to explore the main sources of overall imbalance. Then, we employed MGWR to quantitatively examine the effect of accessibility and institutional service capacity on utilisation and revealed the mechanisms of influence". This has already been said.

Response 16: We have removed these duplicate texts according to the reviewers’ suggestion, in the Conclusion section.

Point 17: The purpose of the study should be at the beginning of the text, not in the Conclusions. Lines 549-551: "The purpose of this study is to identify regions of inequitable utilisation and investigate the major driving factors of it in order to develop more effective positive coping strategies".

Response 17: Thanks for your advice. We have moved this phrase to INTRODUCTION section (page 3 line 103-105) for better expression of our purpose.

Point 18: I recommend reducing the number of bibliographies and citing only those texts that are directly related to the object of the study and that help to understand it better.

Response 18: Thanks for your comments. We have re-simplified the references. Meanwhile, we have retained references directly related to the subject of the study in order to facilitate a better understanding of the text by the reader.

Reviewer 2 Report

This article provides an important insight into access to services in a particularly at-risk population such as the elderly. The study is appropriately conducted with a robust methodology. The clarity of exposition and the transparency with which the research framework was presented is particularly praiseworthy. For this reason, I consider only minor revisions to be necessary:

-        In the abstract, I would state more clearly the aim of the study.

-        I would explain the abbreviation when it first appeared in the main text as well (i.e., ECF, line 47).

-        line 146: how was the weighted centre of the elderly population calculated or what is the data source?

-        line 162: the indicator 'ECF utilization' could be better explained. At what level was it considered, district? sub-district? How was the occupancy estimated? Was the number of occupied beds out of the total number of beds available and if so at what time frame?

-        lines 283-285: the sentence is incomplete.

-        Some sentences of the results section were already in comments to the results so I would move them to the discussion (i.e lines 288-294; lines 314-316; lines 319-320; lines 341-344).

-        The discussion could be improved by comparing accessibility to services with other studies from LMIC countries (doi: 10.7189/jogh.12.04087, PMID: 36273278) where access to NCD services decreased as age increased.

-        The authors state "At the local level, we found that the higher the population density of the region, the lower the walking accessibility". This could be the case that it is not population density that is associated with lower accessibility, but the fact that there is a better public transport network in more densely populated areas (as shown by the reported results in which accessibility is associated with buses and trains in more densely populated areas). For this reason, these means could be preferred over walking.

Author Response

Above all, thanks very much for the comments, which are very helpful to improve the quality of this article. We have revised the manuscript and especially paid much attention to your comments and suggestions.

Point 1: In the abstract, I would state more clearly the aim of the study.

Response 1: We think this is an excellent suggestion. We have revised the abstract to make the aim of the study clearer with sentence “To effectively improve the equity of the service for older adults, a better understanding of how the supply and demand factors affect the actual utilization of regional ECFs is necessary” (page 1 line 14-16).

Point 2: I would explain the abbreviation when it first appeared in the main text as well (i.e., ECF, line 47).

Response 2: We sincerely thank the reviewer for careful reading. We have rechecked and explained all the abbreviations when it first appeared in the main text(page 2 line 48).

Point 3: line 146: how was the weighted centre of the elderly population calculated or what is the data source?

Response 3: The calculation of the weighted centre for the elderly population mainly includes administrative boundary data and elderly population data in sub-districts. The data sources are given in the text as Gaode Map and Chongqing Civil Affairs Bureau. The calculation formula is as follows.

The Mean Center of sub-districts is given as:

where  and  are the coordinates for sub-district i, and n is equal to the total number of sub-districts.

The Weighted Centre of the elderly population is calculated by the following:

where  is the weight of elderly population of sub-district i.

Point 4: line 162: the indicator 'ECF utilization' could be better explained. At what level was it considered, district? sub-district? How was the occupancy estimated? Was the number of occupied beds out of the total number of beds available and if so at what time frame?

Response 4: We thank the reviewer for raising this question. We have explained the indicator "ECF utilization" more clearly in the text for a better understanding(page 4 line 162). In our study, the indicator 'ECF utilization' is considered at the sub-district level. Indeed, the number of occupied beds is out of the total number of available beds. In addition, 'occupancy' means the actual number of elderly people in ECFs. These data come from the 2021 annual statistics of the Chongqing Civil Affairs Bureau for ECFs.

Point 5: lines 283-285: the sentence is incomplete.

Response 5: Thanks for your careful checks. We are sorry for our carelessness. Based on your comments, we have made the correction to make the sentence complete(page 8 line 169).

Point 6: Some sentences of the results section were already in comments to the results so I would move them to the discussion (i.e lines 288-294; lines 314-316; lines 319-320; lines 341-344).

Response 6: Thanks for your suggestion. We have moved these sentences to the DISCUSSION and made cuts to the repetitive points. Your suggestions have optimised the framework and logic of the thesis. These changes are marked in red in the revised paper.

Point 7: The discussion could be improved by comparing accessibility to services with other studies from LMIC countries (doi: 10.7189/jogh.12.04087, PMID: 36273278) where access to NCD services decreased as age increased.

Response 7: We sincerely appreciate the valuable comments. We have added relative references from LMIC countries into the DISCUSSTION part to compare accessibility to services with other studies(page 17 line 502-504).

Point 8: The authors state "At the local level, we found that the higher the population density of the region, the lower the walking accessibility". This could be the case that it is not population density that is associated with lower accessibility, but the fact that there is a better public transport network in more densely populated areas (as shown by the reported results in which accessibility is associated with buses and trains in more densely populated areas). For this reason, these means could be preferred over walking.

Response 8: We are so grateful for your kind question. Based on your suggestions, we have refined the logic of the content in this section. We have explained the causal relationship between population density and travel models to make it more readable and sensible(page 16 line 439-446).
